# Multi-Omics Insights into Rumen Microbiota and Metabolite Interactions Regulating Milk Fat Synthesis in Buffaloes

**DOI:** 10.3390/ani15020248

**Published:** 2025-01-17

**Authors:** Ye Yu, Runqi Fu, Chunjia Jin, Lin Han, Huan Gao, Binlong Fu, Min Qi, Qian Li, Jing Leng

**Affiliations:** 1Faculty of Animal Science and Technology, Yunnan Agricultural University, Kunming 650201, China; yy09091823@163.com (Y.Y.); fandrunqi@163.com (R.F.); jin_chunjia@ynau.edu.cn (C.J.); 19286920591@163.com (L.H.); gaohuanhhhh@163.com (H.G.); binlongfu@126.com (B.F.); qimin538@gmail.com (M.Q.); 13408830996@163.com (Q.L.); 2Key Laboratory of Animal Nutrition and Feed Science of Yunnan Province, Yunnan Agricultural University, Kunming 650201, China; 3Yunnan Animal Husbandry Station, Kunming 650224, China

**Keywords:** rumen microbiome, rumen metabolome, plasma metabolome, milk fat, Binglangjiang buffaloes

## Abstract

The present study was conducted to analyze the effects of milk fat content on the rumen microbiome and metabolomics of Binglangjiang buffaloes (river buffaloes). The results showed that there are differences in the rumen microbial composition and host metabolites of Bingangjiang buffaloes with high and low milk fat contents. These changes in the microbiota may lead to alterations in fatty acid biosynthesis precursors in the rumen and plasma, ultimately affecting milk fat content. Our research revealed a significant impact on these processes, including Quienlla, Stilbella, Cyrenella, Microsphaeropsis, and Paraphaeosphaeria, which are associated with the following metabolites in the rumen: lucidenic acid J, secoeremopetasitolide B, and erinacine P, as well as the plasma metabolite corchorifatty acid F, PC (18:3 (6Z, 9Z, 12Z)/18:3 (9Z, 12Z, 15Z)). The roles of the rumen microbiota and metabolites can be further validated in the future.

## 1. Introduction

Milk is rich in a variety of essential nutrients, including fat, protein, and various trace elements that are important for healthy human development, especially many kinds of unsaturated fatty acids. Milk fat is a milk component with high nutritional value, which is the most important determinant of milk quality. A high fat content is an important characteristic of buffalo milk [1]. In addition, milk fat is an important source of essential fatty acids, phospholipids, and fat-soluble vitamins, providing the body with a large amount of nutrients and energy [2,3]. Buffalo milk, as a best-selling dairy raw material in domestic and international markets, is milky and thick, with a high content of nutrients. Its content of milk fat, milk protein, total solids, and minerals is higher than that of Holstein milk; therefore, the development of buffalo milk is particularly important [4,5].

Ruminants have been domesticated for more than 10,000 years [6] and rely on fermentation by the rumen microbial community to convert complex polysaccharides, which make up a major portion of the plant, into available nutritive substances [7,8]. Furthermore, the rumen microbiota is one of the key factors affecting the formation of milk fat precursors. Changes in milk fat production and milk fatty acid composition are usually caused by alterations in rumen fermentation and biohydrogenation pathways [9]. Previous research has indicated that there are interactions between rumen microbes and the mammary gland. Gastrointestinal microbial metabolites can enter the bloodstream through the intestinal epithelium into the mammary gland, which then influences the composition of milk secreted by the mammary gland [10]. It seems that rumen metabolites, as important precursors for milk fat production, have a direct impact on milk fat percentage in dairy cows [11]. Several studies have shown that the difference in milk fat percentage may be caused by the interactions between gut microorganisms and their metabolites, especially Firmicutes myristic acid and Proteus cholin [12]. However, the levels of metabolites in rumen fluid do not directly reflect the levels of relevant metabolites in the blood [13].

Blood is a commonly used biofluid in metabolomics analyses, providing metabolites from all organs [14]. Blood metabolites can be used as physiological biomarkers to reflect metabolic performance and dysfunction [15]. Meanwhile, some typical metabolites are highly correlated with specific rumen microbiota, suggesting a functional correlation between the microbiome and its associated metabolites [16]. With the development of biotechnology, more researchers are using macro-genomics, macro-transcriptomics, and proteomics to explore the role of the rumen microbiota and their involvement in milk fat synthesis, milk production, and lipid metabolism [12,17]. However, this relationship has not yet been studied in Binglangjiang buffaloes. This buffalo species can be found across all continents and boasts abundant resources. It is classified into two subspecies: river buffaloes (50 chromosomes), primarily inhabiting India, Pakistan, and Mediterranean coastal countries and regions, and swamp buffaloes (48 chromosomes), which are mainly distributed in China and southeast Asian countries [18]. The Binglangjiang buffalo is the only species of river buffalo that has been discovered in China. A previous study found that the fat and protein contents in the milk of Binglangjiang buffaloes was about 2.1 and 1.2 times higher than that of Holstein milk, suggesting that their milk has excellent qualities [19]. However, the individual Binglangjiang buffalo vary greatly in terms of milk composition and reproductive performance, and their daily milk production is still at a low level compared to other countries. In this study, we speculate that the milk fat content of cows fed the same diet may be influenced by rumen microbiota composition and its association with the host metabolism. Therefore, we investigated the correlation between the rumen microbiota and the metabolomics of rumen and plasma in high fat content and low fat content Binglangjiang buffaloes.

## 2. Materials and Methods

### 2.1. Animals and Management

All the experimental protocols were approved by the Animal Care Committee at the Faculty of Animal Science and Technology, Yunnan Agricultural University (Kunming, P. R. China, approval No.: YNAU202205020). In this experiment, we randomly selected 75 healthy mid-lactation Binglangjiang buffaloes with similar lactation days (120–140 days) and similar body condition (body condition score from 2.5 to 3). Then, they were kept in a commercial buffalo farm in Tengchong City, Yunnan Province. The buffaloes were kept in individually tethered stalls in a barn with good ventilation and were fed 2 times daily at 08:00 h and 16:00 h. The buffaloes were fed the same diet (Appendix A) with a concentrate-to-forage ratio of 35:65 (DM basis) and had free access to water. Then, we continuously measured the composition of the milk for 6 weeks once a week, and the 10 highest milk fat content buffaloes (HF, 5.60 ± 0.61%) and 10 lowest milk fat content buffaloes (LF, 1.49 ± 0.13%) were selected accordingly (Appendix A).

### 2.2. Sample Collection and Measurement

Milk samples were collected at 06:30 with a bronopol tablet (milk preservative, D & F Control Systems, San Ramon, CA) and then stored at 4 °C before the infrared analysis of milk components and somatic cells using a spectrophotometer (Foss-4000, Foss, Hillerød, Denmark). Milk production was recorded for 7 consecutive days using a mobile milking machine, and feed intake was recorded prior to formal sample collection. Blood samples were collected from the jugular vein, and plasma was separated and then stored at −80 °C for metabolomic analysis. Rumen contents were sampled using oral stomach tubes and were used to measure microbiota and metabolites. The milk, blood and rumen samples were collected on the same day after milk production was recorded. The pretreatment and determination of rumen fluid VFAs were carried out with reference to Hu et al. [20].

### 2.3. 16S rDNA and ITS Sequencing and Data Processing

Rumen microorganisms DNA were extracted from 2 mL rumen fluid by the kit method (Omega Bio-tek, Norcross, GA, USA). Before PCR amplification, we used 1.0% agarose gel electrophoresis and NanoDrop2000 to detect the purity and concentration of DNA, and we took an appropriate amount of sample from the centrifuge tube for dilution. Diluted DNA was used as a template. Specific primers were selected to amplify the V3 + V4 region of 16SrDNA. The fungus was subjected to ITS sequencing, and the amplified region was ITS1F-ITS2R. The primer sequences and PCR amplification conditions were referred to Yu et al. [21].

We used FASTP (v0.20.0) software to control the quality of the original sequencing sequence [22] and used FLASH (v1.2.7) software to splice [23]. Using UPARSE software (v7.1), OTU clustering was performed on sequences according to 97% similarity, and chimeras were eliminated [24]. RDP classifier (v2.2) was used to classify and annotate each sequence [25], compared with the Silva 16S rDNA database (v138), and we set the comparison threshold to 70%. Then, we used Mothur software (v1.30.2) for alpha diversity analysis. Beta diversity was analyzed by principal component analysis (PCA) based on the Bray–Curtis distance in R language (v 3.3.1), while the 16S and ITS sequence data were compared between the two groups (HF vs. LF) using the Wilcoxon rank-sum test with the *p*-value < 0.05 considered significantly different.

### 2.4. Analysis of Rumen and Plasma Metabolome

Rumen and plasma metabolites were extracted according to Guo et al. [26]. First, 100 μL liquid sample was added to a 1.5 mL centrifuge tube with 400 μL solution (acetonitrile: methanol = 1:1 (v:v)) containing 0.02 mg/mL internal standard (L-2-chlorophenylalanine) to extract metabolites. The samples were mixed by vortex for 30 s and low-temperature sonicated for 30 min (5 °C, 40 KHz). The samples were placed at −20 °C for 30 min to precipitate the proteins. Then, the samples were centrifuged for 15 min (4 °C, 13,000× *g*). The supernatant was removed and blown dry under nitrogen. The sample was then re-solubilized with 100 µL solution (acetonitrile: water = 1:1) and extracted by low-temperature ultrasonication for 5 min (5 °C, 40 KHz), which was followed by centrifugation at 13,000× *g* and 4 °C for 10 min. The supernatant was transferred to sample vials for LC-MS/MS analysis.

As a part of the system conditioning and quality control process, a pooled quality control sample (QC) was prepared by mixing equal volumes of all samples. The LC-MS analysis was performed on the Thermo UHPLC-Q system of Majorbio bio-pharma Technology Co. (Shanghai, China). A Thermo UHPLC-Q Exactive HF-X mass spectrometer was used to collect mass spectrum data, which was divided into two working modes: positive mode and negative mode. The data-related acquisition (DDA) mode was used for data acquisition. It was tested in the quality range of 70–1050 m/z. The LC/MS raw data were pre-processed using Progenesis QI (Waters Corporation, Milford, CT, USA) software (version 2.0). At the same time, we used the metabolite identification of a searchable database, which was the main database for the KEGG database (http://www.genome.jp/kegg/, accessed: 20 February 2023).

We uploaded a searchable database of the data matrix to the Majorbio cloud platform (https://cloud.majorbio.com, accessed: 15 March 2020) for data analysis. First, the data matrix was preprocessed: at least 80% of the metabolic signatures detected in each set of samples were retained. After filtration, minimum metabolite values were estimated for specific samples whose metabolite levels were below the lower quantization limit, and various metabolic characteristics were normalized to sum. The response intensity of the sample quality spectrum peak was normalized by the sum normalization method. At the same time, the variables of QC samples with a relative standard deviation (RSD) of >30% were removed, and log10 was used for subsequent analysis. Then, the R software package “ropls” (version 1.6.2) was used for orthogonal least partial binary discriminant analysis (OPLS-DA) and interactive verification to evaluate the stability of the model. According to the *p*-values generated by the OPLS-DA model and Student’s t-test, metabolites with *p* < 0.05 were significantly different metabolites.

Differential metabolites among the two groups were mapped into their biochemical pathways through metabolic enrichment and pathway analysis based on the KEGG database. Python packages “scipy.stats” (https://docs.scipy.org/doc/scipy/, accessed: 27 March 2023) was used to perform enrichment analysis to obtain the most relevant biological pathways for experimental treatments. Correlation analysis between the rumen microbiota, rumen and plasma metabolome were performed using the Spearman (https://cloud.majorbio.com/page/tools.html, accessed: 5 April 2023), any *p*-value below 0.05 was regarded as indicating a significant correlation.

## 3. Results

### 3.1. Characterization of Phenotypes

In this study, 10 buffaloes with the highest milk fat content (HF) and 10 buffaloes with the lowest milk fat content (LF) were selected for 16S, ITS, rumen metabolome, and plasma metabolome analyses. Among the phenotypes, the milk fat and lactose content was significantly different between the HF and LF (*p* < 0.001) (Table 1). Concentrations of rumen volatile fatty acids were quantified and found to be significantly higher (*p* < 0.05) in HF buffaloes for total volatile fatty acids (TVFAs) and acetate (Appendix A).

### 3.2. Rumen Bacteria and Taxonomic Differences Between the HF and LF Buffaloes

In 16S rDNA sequencing, a total of 960,813 sequences were generated, with 48,040 ±1523 sequences (mean ± standard error of the mean [SEM]) per sample, and the average sequences in group HF and LF were 48,050 and 48,031, respectively (Appendix A). The Chao, Shannon and coverage index of HF and LF buffaloes were not significantly different (Figure 1A–C). The partial least squares discriminant analysis (PLS-DA) showed separations between the groups based on bacterial phyla (Figure 1D) and genus (Figure 1E).

The dominant bacterial phyla included Bacteroidetes (55.98 ± 1.02%), Firmicutes (27.32 ± 1.14%), and Proteobacteria (7.32 ± 1.57%); the dominant bacterial genus was *Prevotella* (21.20 ± 0.45%), which was followed by *Rikenellaceae_ RC9_ gut_ group* (10.05 ± 0.05%), *norank_f__UCG-011* (6.34 ±0.49%) and *Succiniclasticum* (4.20 ± 0.34%) (*p* < 0.05, Figure 1F,G). For differential abundance comparison analysis at the phylum level, the abundance of Synergistota was significantly higher in the rumen of HF buffaloes. At the genus level, *Quinella*, *Selenomonas* and *Fretibacterium* exhibited significantly higher abundances in the rumen of HF buffaloes (*p* < 0.05), while 12 genera showed significant enrichment in the rumen of LF buffaloes (*p* < 0.05; Figure 2).

### 3.3. Rumen Fungi and Taxonomic Differences Between the HF and LF Buffaloes

In ITS sequencing, a total of 1,209,963 sequences were generated, with 60,498 ±2540 sequences (mean ± standard error of the mean [SEM]) per sample, and the average sequences in the HF and LF groups were 62,041 and 58,965, respectively (Appendix A). The Chao, Shannon and coverage indexes of HF and LF buffaloes were not significantly different (Figure 3A–C). The partial least squares discriminant analysis (PLS-DA) showed separations between the groups based on fungi phyla (Figure 3D) and genus (Figure 3E).

The dominant fungal phyla included Ascomycota (55.98 ± 1.02%), Basidiomycota (27.32 ± 1.14%), and Neocallimastigomycota (7.32 ± 1.57%); the dominant genus was *Orpinomyces* (14.83 ± 0.11%), which was followed by *Cutaneotrichosporon* (8.32 ± 0.41%), *norank_f_UCG-011* (6.34 ± 0.49%) and *Trichosporon* (7.49 ± 2.90%) (Figure 3F,G). For differential abundance comparison analysis at the phylum level, there was no significant difference between HF and LF buffaloes (*p* > 0.05). At the genus level, the abundance of 14 rumen fungi, included *Candida*, *Talaromyces*, *Cyrenella* and *Stilbella*, was significantly higher in the rumen of HF buffaloes (*p* < 0.05), while *Anaeromyces* showed significant enrichment in the rumen of LF buffaloes (*p* < 0.05; Figure 4).

### 3.4. Rumen Metabolome

To discriminate the metabolic profiles across groups, we performed clustering analyses based on orthogonal partial least square discriminant analysis (OPLS-DA). The rumen samples from distinct groups were largely separated according to the OPLS-DA plots (Figure 5A). A total of 1451 compounds were identified in the rumen metabolome, and nine of these metabolites were identified in the HF group only, e.g., PE (15:0/16:0), LPC (18:1), and LPC (18:2) (Appendix A). After t-test filtering for the relative concentrations of rumen metabolites, 68 metabolites were significantly different between the two groups, 40 of which were significantly higher in the rumen of HF buffaloes (Figure 5B, Appendix A). By clustering these differential metabolites, most of those clustered in the HF group were lipid and lipid-like molecules such as secoeremopetasitolide B and lucidenic acid J (Figure 5C,D). KEGG pathway enrichment analysis based on these 68 significantly different rumen metabolites revealed the enrichment of five pathways, which included “Bile secretion”, “Drug metabolism-cytochrome P450”, “Arginine and proline metabolism”, “Pentose and glucuronate interconversions”, and “ Phenylalanine metabolism”; among these, the “Pentose and glucuronate interconversions” pathway was significantly upregulated in the HF group (*p* < 0.05, DA score > 0.1) (Figure 5E).

### 3.5. Plasma Metabolome

Similar to the rumen, to discriminate the metabolic profiles across groups, we performed clustering analyses based on orthogonal partial least square discriminant analysis (OPLS-DA). The rumen samples from distinct groups were largely separated according to the OPLS-DA plots (Figure 6A). A total of 770 compounds were identified in the plasma metabolome; among these, three of the metabolites were identified in the HF group only, namely alpha-crocetin glucosyl ester, 3,4,5-trihydroxy-6-[(3-oxo-1,7-diphenylheptan-2-yl)oxy]oxane-2-carboxylic acid, and isovalerylglutamic acid (Appendix A). After t-test filtering for the relative concentrations of plasma metabolites, 42 metabolites were significantly different between the two groups, 36 of which were significantly higher in the rumen of HF buffaloes (Figure 6B, Appendix A). By clustering these differential metabolites, most of those clustered in the HF group were lipid and lipid-like molecules such as LysoPE (0:0/18:2 (9Z, 12Z)), 5-tetradecenoic acid and hexadecanedioic acid (Figure 6C,D). KEGG pathway enrichment analysis based on these 42 significantly different plasma metabolites revealed the enrichment of six pathways, which included “Pyrimidine metabolism”, “Lysine degradation”, “Pantothenate and CoA biosynthesis”, “Choline metabolism in cancer” and “Glycerophospholipid metabolism”; among these, the “Pantothenate and CoA biosynthesis” pathway was significantly upregulated in the HF buffaloes (*p* < 0.05, DA score > 0.1) (Figure 6E).

### 3.6. Relationships Between the Rumen Microbiome, Rumen Metabolome and Plasma Metabolome

Spearman’s rank correlations between the rumen microbiota and rumen metabolites were assessed. The results showed that *Quinella*, *Fretibacterium*, *Selenomonas*, *Cyrenella*, *Stilbella* and *Candida* were significantly enriched in HF buffaloes and showed a significant positive correlation with rumen metabolites such as lucidenic acid J, secoeremopetasitolide B and erinacine P (*p* < 0.05). Meanwhile, they were negatively correlated with organic compounds such as 3-amino-2-naphthoic acid, N-acetyl-b-glucosaminylamine, and flazine (*p* < 0.05) (Figure 7A,B). Similarly, Spearman’s rank correlations between the rumen microbiota and plasma metabolites showed that *Quinella*, *Fretibacterium* and *Selenomonas* were significantly enriched in HF buffaloes and showed significant positive correlation with plasma metabolites such as corchorifatty acid F, PC (18:3 (6Z, 9Z, 12Z)/18:3(9Z, 12Z, 15Z)), hexadecanedioic acid, LysoPE (0:0/18:2 (9Z, 12Z)) and 5-tetradecenoic acid (*p* < 0.05). The rumen fungi such as *Cyrenella*, *Stilbella* and *Beauveria* showed significant positive correlations with dehydroadoreone, 3′,4′,5′-trimethoxycinnamyl alcohol acetate and prostaglandin E1. Meanwhile, *Candida*, *Phaeosphaeriopsis* and *Microsphaeropsis* showed significant negative correlations with LysoPC (22:4 (7Z, 10Z, 13Z, 16Z)) (Figure 7C,D). Spearman’s rank correlations between the differential rumen microbiota and rumen-specific metabolites were performed. The results showed that *Quinella*, *Fretibacterium* and *Selenomonas* were significantly enriched in the HF group and showed significant positive correlations with metabolites such as Bisnorbiotin, PE (15:0/16:0), LPC (18:1), and isovalerylglutamic acid (*p* < 0.05). Then, *Paeniclostridium*, *Lachnoclostridium*, and *Monogobus* were significantly and negatively correlated with isovalerylglutamic acid (*p* < 0.05) (Appendix A).

To identify the rumen microbiota–metabolic interactions, Spearman’s rank correlations between the rumen microbiota, rumen and plasma metabolites were performed (R > 0.7/R < –0.7, *p* < 0.05). Among these correlations, *Quienlla*, *Stilbella*, *Cyrenella*, *Microsphaeropsis*, and *Paraphaeosphaeria* showed correlations with rumen and plasma metabolites, which included lucidenic acid J, N-acetyl-b-glucosaminylamine, flazine, corchorifatty acid F, LysoPE (16:1 (9Z)/0:0), hexadecanedioic acid, isovalerylglutamic acid, PC (18:3 (6Z, 9Z, 12Z)/18:3 (9Z, 12Z, 15Z)), and other metabolites (Figure 8).

## 4. Discussion

The dominant rumen microbiota plays an important role in the lactation performance of the host. Current studies have shown that the dominant phyla of cows’ rumen microorganisms were Bacteroidetes, Firmicutes and Proteobacteria [26], which is consistent with our study using 16S rDNA gene amplicon sequencing. Among these, Bacteroidetes mainly decompose non-fiber carbohydrates, while Firmicutes mainly decompose fibers, and there was a strong positive correlation between the milk fat yield and the proportion of Firmicute and Bacteroidetes [27]. The proportions of the Firmicutes and Bacteroidetes in the rumen of high yielding cows were significantly higher compared with low yielding cows [28]. Meanwhile, changes in diet or feeding preferences can also lead to changes in rumen microbiota, which in turn affect lactation performance. Exploring the mechanisms of rumen microbial metabolism of carbohydrates, proteins and sugars can be targeted to regulate rumen microbial fermentation and promote the production of milk component precursors through diets.

The relative abundance of some rumen microbiota has been found to have an effect on milk yield and milk fat content [29]. Furthermore, some studies have shown that rumen bacterial abundance was correlated with milk fat or protein in dairy cows [30,31]. In this study, the abundance of *Quinella*, *Selenomonas* and *Fretibacterium* was significantly higher in the HF buffaloes at the genus level. *Quinella* and *Selenomonas* all belong to the phylum Firmicutes [32,33]. A higher relative ruminal propionate concentration was also found by Kittelmann et al. (2014) in sheep with elevated populations of *Quinella*; it was hypothesized that *Quinella* would be conducive to lower methane production, and earlier studies have also supported that [34,35]. Further research found that *Quinella* genomic bins contained genes that code for pyruvate: ferredoxin oxidoreductase, which converts pyruvate to acetyl-CoA, and *Quinella* may be able to utilize cellulose and cellulodextrins released by other rumen microbiota using β-glucosidase [36]. In addition, β-xylosidase from *Selenomonas* has high β-xylosidase activity [37]. There are also studies that have found *Schwartzia* is a genus in the Firmicutes, which was more abundant in cows with higher milk production [27]. Li et al. (2020) found that Lachnoclostridium and Lachnospiraceae UCG-006 could inhibit short-chain fatty acid producing bacteria and thus inhibit the synthesis of milk fat, which was also confirmed by the significant enrichment of Lachnoclostridium in LF buffaloes in this study [38]. Milk fat contents have also been correlated with the abundance of Dialister, Megasphaera, Lachnospira, and Sharpea in the rumen [39]. Then, another study has shown that the proportion of *Prevotella* in the Bacteroidetes and the milk fat yield were significantly negatively correlated [27,40]. The relative abundance of rumen *Prevotella* increased significantly when cows developed low milk syndrome, while the relative abundance of *unclassified_Lachnospiraceae*, *unclassified_Veillonellaceae*, and *Pseudobutyri-vibrio* decreased significantly [41]. Additionally, plant cell wall polysaccharide-degrading enzymes expressed by rumen microbiota were key in regulating the production of milk fat synthesis precursors, suggesting that HF buffaloes may be producing more milk fat synthesis precursors such as volatile fatty acids. Notably, fewer studies have been conducted to correlate the effects of rumen fungi on milk fat synthesis, but that does not mean it is not important. In this study, the abundance of 14 rumen fungi, including *Candida*, *Talaromyces*, *Cyrenella* and *Stilbella*, was significantly higher in the rumen of HF buffaloes. This suggested that rumen fungi similarly play an important role in milk fat anabolism, which may require further investigation in the future. With the continuous development of sequencing technology, researchers have studied the diversity of rumen microorganisms and their enzyme genes using techniques such as macrogenomics, macrotranscriptomics, and macroproteomics. It opens the way to the discovery of more functional microorganisms and enzyme genes in the future, but there are still a large number of unexploited microorganisms in the rumen, and their isolation and expression in engineered bacteria will be promising for improving milk fat synthesis.

Alterations in the gastrointestinal microbiota affect metabolites [42,43] as well as metabolic pathways in dairy cows [44]. Ruminal microorganisms produced a large number of small-molecule metabolites during digestion in dairy cows, and these metabolites not only play a key role in colony messaging but also alter the metabolic state of their hosts [45,46]. Rumen metabolomics studies have shown that phospholipids, amino acids, fatty acids, carbohydrates, cholesterol esters and glycerides are the major metabolites in the rumen [41]. In the present study, it was found that the major differential metabolites in the rumen of high milk fat cows were lipid and lipid-like molecules such as triglycerides and phosphatidylethanolamine, which is consistent with the results of previous studies [31]. The synergistic effect of rumen microbes and their metabolites in dairy cows can increase the gene expression of hemicellulases, lipid synthases and transferase enzymes, which in turn increases the rumen’s ability to break down nutrients from the ration and ultimately promotes an increase in milk fat content [47]. *Lactobacilli*, *Ruminalococcus* and *Clostridia* in the phylum Firmicutes have bile salt hydrolase activity, which produces short fatty acids, lactate and antimicrobial substances involved in the regulation of traits such as lactation organisms and immune responses [48,49]. Another study found that bacteria of the *Prevotella* and *Succinomonas* in the rumen of dairy cows could affect the concentration of glutathione and phenylalanine in the rumen, which in turn affected the concentration of amino acids, such as glycine, serine, and threonine, in the serum of dairy cows, and regulated the production of milk lipids in cows [29]. It has been found that bacterial metabolites of Bacteroidetes could regulate the gluconeogenesis pathway by modulating bile acids, i.e., promoting cholesterol efflux through the dissociation of bile salts and thus regulating the concentration of triacylglycerol in the blood [50,51]. In this study, it was found that Synergistota was significantly enriched in the HF group. Bacteroidetes mainly decomposes non-fiber carbohydrates, which in turn increases the concentration of acetic acid in the rumen and promotes the synthesis of milk fat, which may be one of the reasons for the high milk fat content in the HF group, and the specific mechanism needs to be further explored. *Eubacterium_xylanophium_group* and *Clostridium* and *Ruminalococcus* play an important role in the fermentation of cellulose-rich feeds and resistant starch and can produce acetic and butyric acids to increase the production of volatile fatty acids, which in turn promotes the synthesis of milk fat [52,53]. Furthermore, some of the lipopolysaccharides produced by rumen microorganisms can damage the blood–milk barrier and then enter the mammary epithelial cells of dairy cows through the blood circulation to cause mastitis, leading to a decrease in milk fat [54]. Therefore, rumen metabolites may affect milk lipogenesis through immune responses and other metabolic pathways. However, milk fat synthesis is a complex process, and the molecular mechanisms of regulation need to be further investigated.

Changes in gastrointestinal microorganisms largely affect the metabolites in the blood [55]. Blood metabolome analysis showed that the high-fat content group enriched in carbohydrates had much higher levels of citric and succinic acids in their serum than the low-fat group, confirming that the tricarboxylic acid cycle is more active in the high-fat group, and pyruvic acid can be converted to acetyl coenzyme A by oxidative decarboxylation [56]. Acetyl coenzyme A can participate in the tricarboxylic acid cycle to produce adenosine triphosphate (ATP) to meet the higher energy requirements of cows with high milk fat content [56]. Plasma concentrations of phosphatidylcholine were found to be higher in cows with a high milk fat content than in cows with a low milk fat content, which were absorbed by the mammary gland for triglyceride synthesis [57]. Lipid-related metabolites were increased in the group of cows with high milk fat content, whereas amino acid metabolism was enhanced and fatty acid metabolism was blocked in cows under conditions of reduced milk fat [41]. In the present study, lipid and lipid-like molecules such as LysoPE (0:0/18:2 (9Z, 12Z)) were found to be significantly upregulated in the HF group. Yamamoto et al. found that lysoPE 18:2 could be involved in the formation of lipid droplets and increased the milk fat content [58]. It has recently been shown that cows at peak lactation have higher LysoPE (16:0) in the rumen, which increases phospholipid levels in milk [59]. Metabolites that differed between groups in plasma samples from cows with different milk fat content were mainly enriched in the choline metabolism, glycerophospholipid metabolism, and amino acid metabolism [13], which is consistent with the results of this study. On the other hand, milk fat content was positively correlated with the lipopolysaccharide biosynthesis pathway, amino acid biosynthesis pathway, valine biosynthesis pathway, leucine biosynthesis pathway, and isoleucine biosynthesis pathway [60]. Meanwhile, the relationship between milk fat content and microorganisms and metabolites is complex and diverse. Our current study demonstrates a positive correlation between specific microbiota and both rumen metabolites and plasma metabolites. The rumen microbiota such as *Quienlla*, *Stilbella*, *Cyrenella*, *Microsphaeropsis*, and *Paraphaeosphaeria* showed correlations with rumen and plasma metabolites, including lucidenic acid J, N-acetyl-b-glucosaminylamine, flazine, corchorifatty acid F, and LysoPE (16:1 (9Z)/0:0). Spearman’s correlation results showed that the metabolites that positively correlated with milk fat content were mainly lipids and organic acids, suggesting that these metabolites provide more milk fat precursors and metabolic energy to the mammary glands of the HF buffaloes via the bloodstream. Additionally, the phenotype-associated metabolites in both the rumen and serum were also significantly correlated with the rumen microbiota, suggesting a responsive relationship between these metabolites and the rumen microbiota [31]. In summary, the composition of rumen microbiota affected both microbial and host metabolism, which, in turn, impacted the host milk fat production traits. However, milk fat synthesis is the result of a concerted effort between gastrointestinal microbes and the organism. Not only do rumen microbes and host metabolites have an impact on milk lipid synthesis, but mammary lactation genes and signaling pathways also play a key role in the regulation of lipid synthesis. Therefore, it is necessary to further investigate the regulatory mechanism of milk fat synthesis with the help of genomics and the transcriptome, which is of great scientific significance, as it can help make full use of the lactation potential of buffaloes and produce high-quality milk.

## 5. Conclusions

There are differences in the rumen microbial composition and host metabolites of Bingangjiang buffaloes with high and low milk fat content. These changes of microbiota may lead to alterations in fatty acids biosynthesis precursors in the rumen and plasma, and it ultimately affected milk fat content. Our findings associated *Quienlla*, *Stilbella*, *Cyrenella*, *Microsphaeropsis* and *Paraphaeosphaeria* with the rumen metabolites lucidenic acid J, secoeremopetasitolide B and erinacine P and the plasma metabolites corchorifatty acid F and PC (18:3 (6Z, 9Z, 12Z)/18:3 (9Z, 12Z, 15Z)), which may be key contributors to high milk fat content in HF buffaloes. The role of these microorganisms can be further validated in the future. Moreover, the mechanism of formation of buffalo milk fat content at the organizational and cellular levels requires further investigation.

## Figures and Tables

**Figure 1 animals-15-00248-f001:**
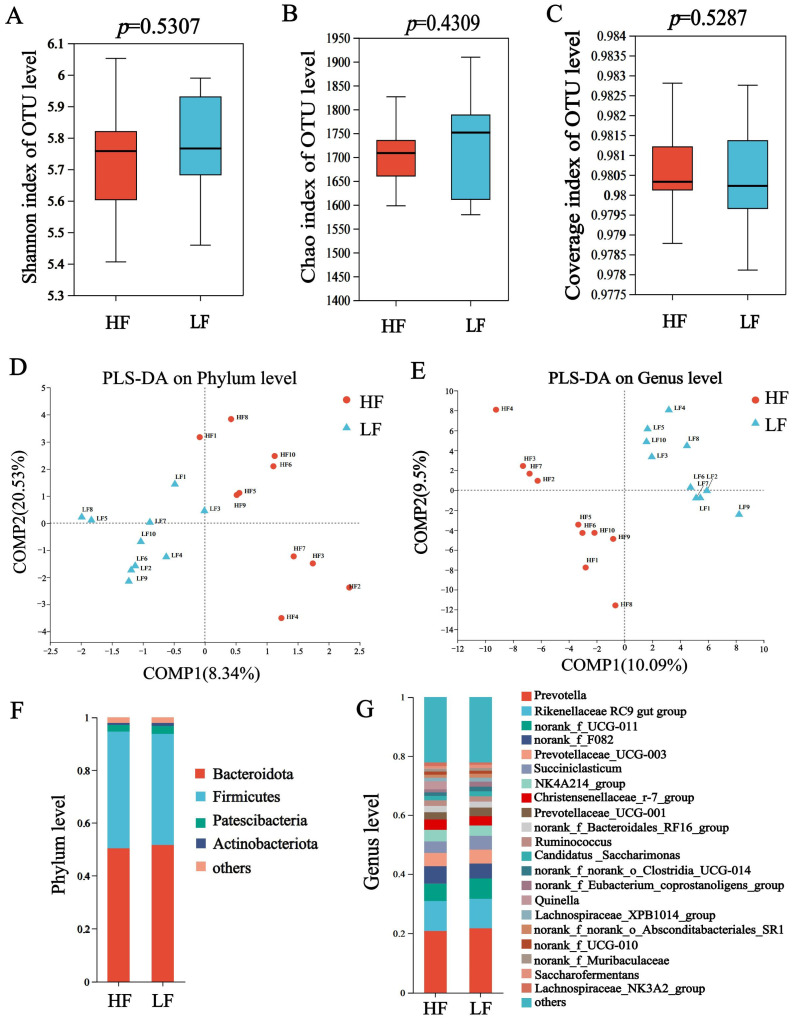
Composition and differential bacteria in the rumen. (**A**) Shannon index. (**B**) Chao index. (**C**) Coverage index. (**D**) The partial least squares discriminant analysis at phylum level. (**E**) The partial least squares discriminant analysis at genus level. (**F**) Community structure at the phylum level. (**G**) Community structure at the genus level.

**Figure 2 animals-15-00248-f002:**
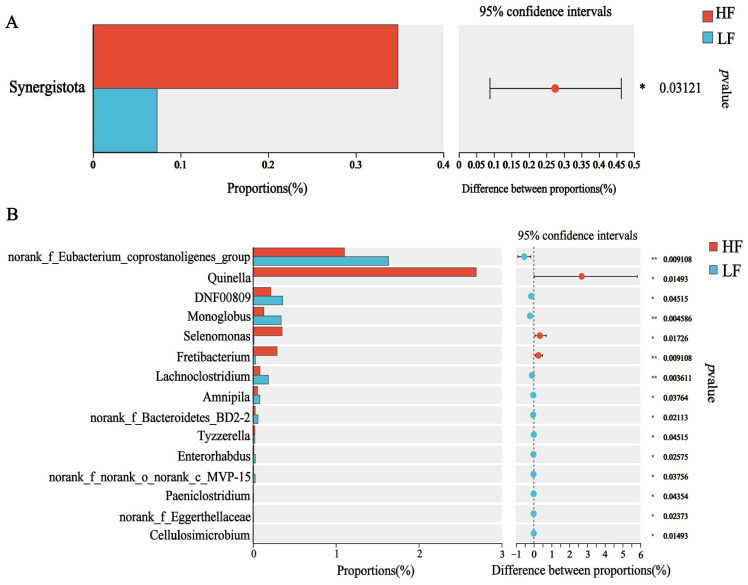
Differential bacteria. * means *p* < 0.05, ** means *p* < 0.01. (**A**) At phylum level. (**B**) At genus level.

**Figure 3 animals-15-00248-f003:**
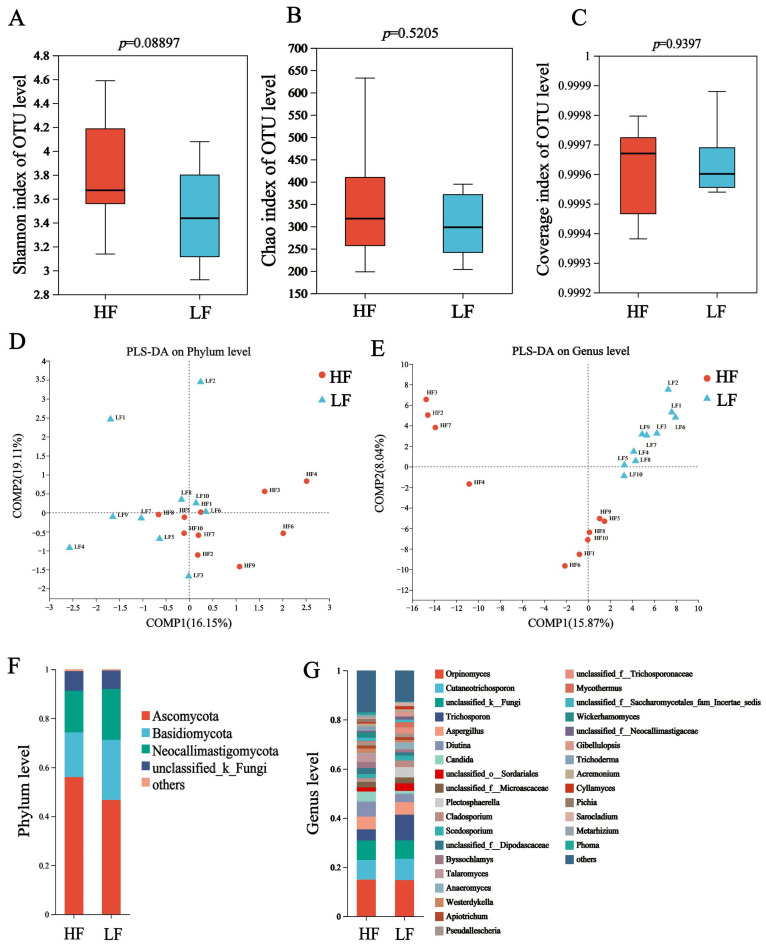
Composition and differential fungi the rumen. (**A**) Shannon index. (**B**) Chao index. (**C**) Coverage index. (**D**) The partial least squares discriminant analysis at phylum level. (**E**) The partial least squares discriminant analysis at genus level. (**F**) Community structure at the phylum level. (**G**) Community structure at the genus level.

**Figure 4 animals-15-00248-f004:**
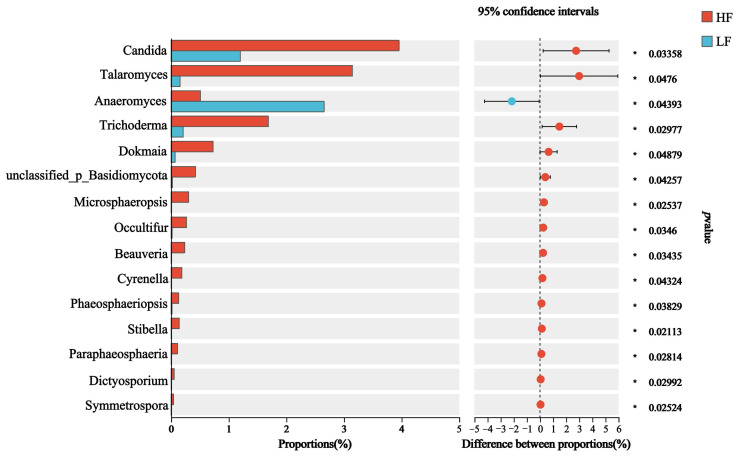
Differential fungi at genus level. * means *p* < 0.05.

**Figure 5 animals-15-00248-f005:**
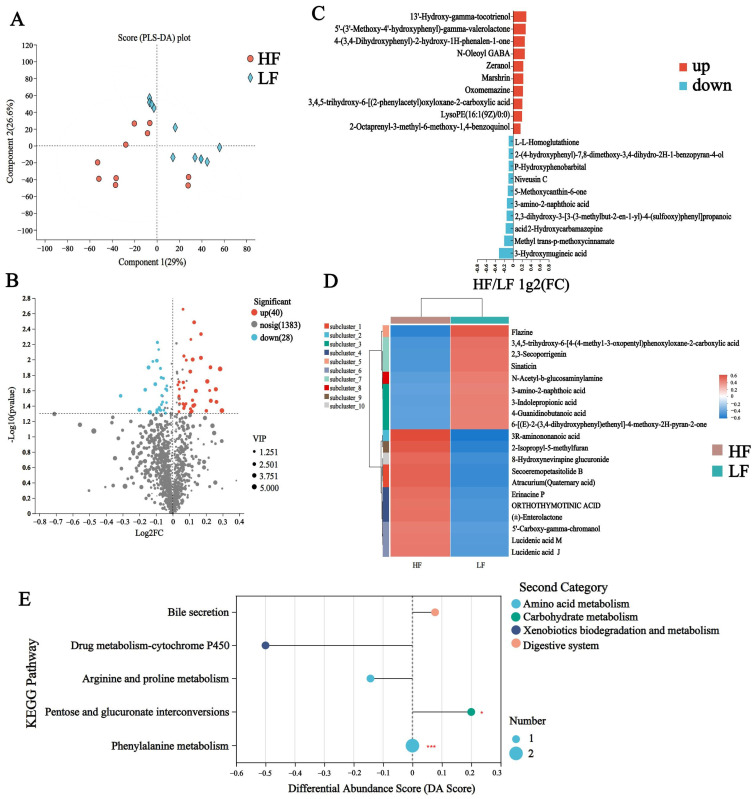
Rumen metabolome of HF and LF buffaloes. (**A**) PLS-DA analysis of rumen metabolome. (**B**) Volcanogram of rumen differential metabolome. (**C**) Comparison of rumen metabolome between HF and LF buffaloes visualized. (**D**) Cluster analysis of differential rumen metabolites, the left is a dendrogram of metabolite clustering, on the right is the name of the metabolite, and the closer two metabolite branches are to each other, the closer they are in terms of expression. (**E**) Differential abundance score map of KEGG pathway of rumen metabolome. A positive DA score indicates that the expression trend of all annotated differential metabolites in the pathway was upregulated in the HF group, the dots are distributed to the right of the center axis and the longer the line segment, the more the overall expression of the pathway tends to be upregulated. * means *p* < 0.05, *** means *p* < 0.001.

**Figure 6 animals-15-00248-f006:**
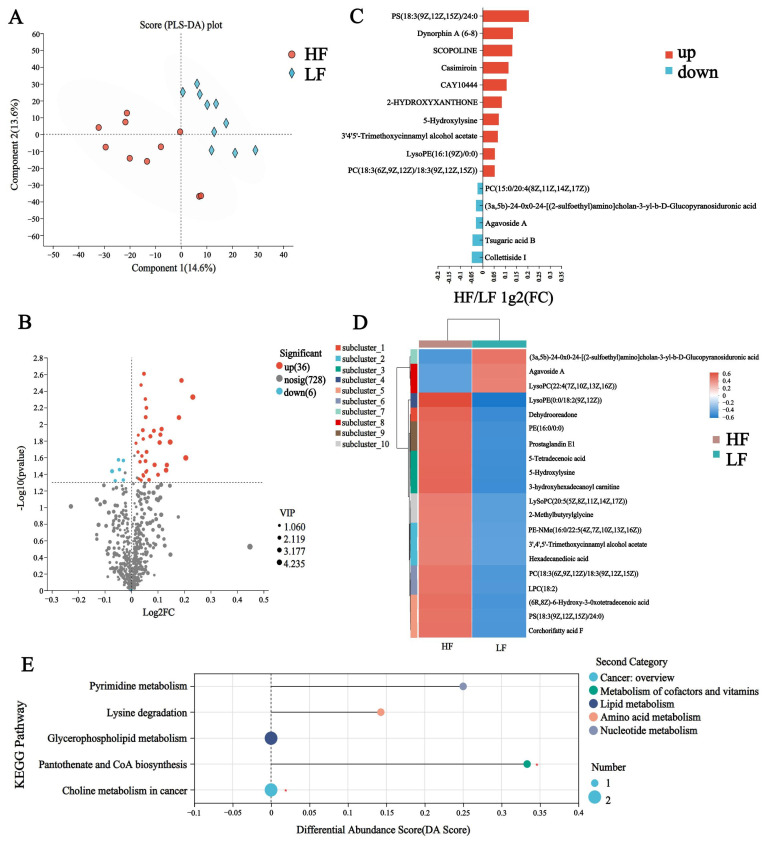
Plasma metabolome of HF and LF buffaloes. (**A**) PLS-DA analysis of plasma metabolome. (**B**) Volcanogram of plasma differential metabolome. (**C**) Comparison of plasma metabolome between HF and LF buffaloes visualized. (**D**) Cluster analysis of differential plasma metabolites. On the left is a dendrogram of metabolite clustering, on the right is the name of the metabolite, and the closer two metabolite branches are to each other, the closer they are in terms of expression. (**E**) Differential abundance score map of KEGG pathway of plasma metabolome. A positive DA score indicates that the expression trend of all annotated differential metabolites in the pathway was upregulated in the HF group; the dots are distributed to the right of the center axis, and the longer the line segment, the more the overall expression of the pathway tends to be upregulated. * means *p* < 0.05.

**Figure 7 animals-15-00248-f007:**
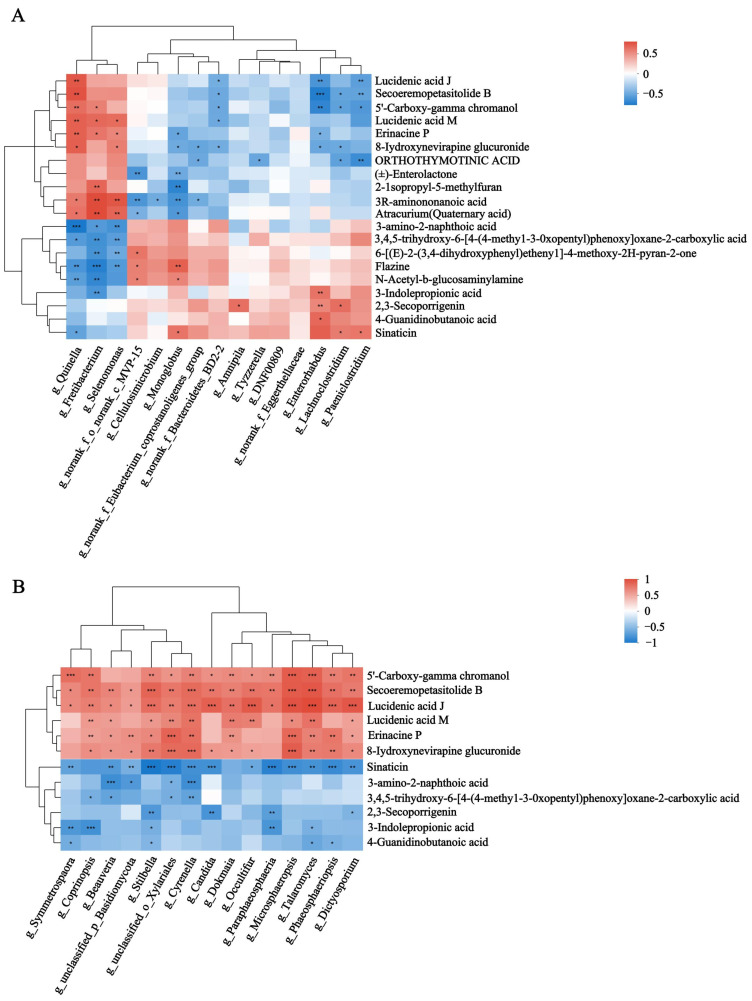
Spearman’s rank correlations between the rumen microbiota and metabolites. (**A**) Differential bacteria and rumen differential metabolites. (**B**) Differential fungi and rumen differential metabolites. (**C**) Differential bacteria and plasma differential metabolites. (**D**) Differential fungi and plasma differential metabolites. * means *p* < 0.05, ** means *p* ≤ 0.01, *** means *p* ≤ 0.001.

**Figure 8 animals-15-00248-f008:**
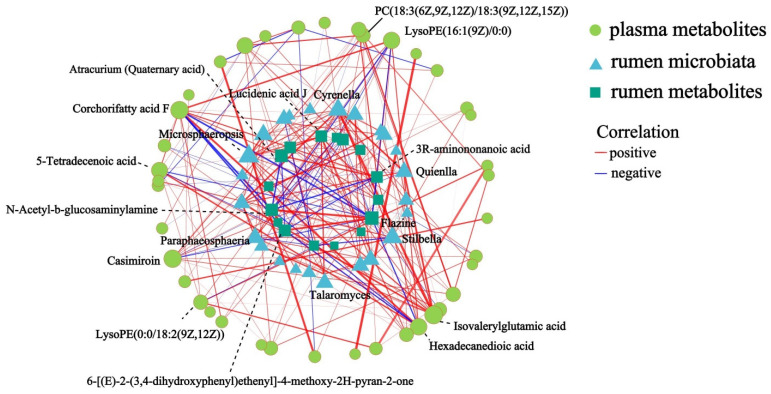
Network analysis of rumen microbiota, rumen metabolites and plasma metabolites (R > 0.7/R < −0.7, *p* < 0.05). The size of the node in the graph indicates the degree size of the node, different colors indicate different species; the color of the connecting line indicates positive and negative correlation, red indicates positive correlation, blue indicates negative correlation; the thickness of the line indicates the size of the correlation coefficient, the thicker the line, the higher the correlation between the species; the shorter the line, the closer the connection between the node.

**Table 1 animals-15-00248-t001:** Physiological parameters of HF and LF buffaloes.

Items	Mean ± SEM		*p*
	HF (*n* = 10)	LF (*n* = 10)	
Milk yield (kg/d)	8.05 ± 0.31	7.03 ± 0.37	0.81
Milk fat (%)	5.60 ± 0.61	1.49 ± 0.13	<0.001
Milk protein (%)	4.85 ± 0.21	4.42 ± 0.10	0.076
Lactose (%)	5.24 ± 0.06	5.57 ± 0.08	0.003
Parity	2.10 ± 0.18	2.50 ± 0.17	0.12
Days in milk (DIM, d)	182.30 ± 13.36	170.40 ± 21.97	0.65
Body weight (kg)	505.00 ± 6.56	508.73 ± 7.02	0.37
Age	4.90 ± 0.23	5.10 ± 0.28	0.60
Dry matter intake (DMI, kg)	10.11 ± 0.16	9.76 ± 0.21	0.21

## Data Availability

The datasets presented in this study can be found in online repositories. The names of the repository/repositories and accession number(s) can be found at: NCBI–PRJNA1205516.

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
