# Peer review of "Multi-Omics Insights into Rumen Microbiota and Metabolite Interactions Regulating Milk Fat Synthesis in Buffaloes"

_animals, 2025, doi:10.3390/ani15020248_

Round 1

Reviewer 1 Report (Previous Reviewer 1)

Comments and Suggestions for Authors

Dear Authors,

After the modifications performed, your manuscript might be now suitable for publication in Animals after minor revision. Please, see below a list of comments/suggestions to be also applied by you before accepting it.

Yours sincerely,

Reviewer.

Check hyphoning throughout the manuscript.

Check grammar in Figures 1 and 3.

Increase full size to gain visibility in Figures 3G, 5C-5D and 6C-6D.

Author Response

Response to Reviewers Comments

Thank you very much for taking the time to review this manuscript. On behalf of all the contributing authors, I would like to express our deepest respect and gratitude to the editor and reviewers for your hard work! As the reviewer was concerned, there are several issues that need to be addressed. We tried our best to improve the manuscript and made some changes to the manuscript, the detailed corrections are listed below and marked in red in the revised manuscript. We are very grateful to the reviewer for their enthusiastic work and hope that the correction will meet with approval.

Comment: After the modifications performed, your manuscript might be now suitable for publication in Animals after minor revision. Please, see below a list of comments/suggestions to be also applied by you before accepting it.

Response: We sincerely appreciate the valuable comments and constructive suggestions. We have carefully revised the article based on the reviewers' comments.

Point-by-point response to Comments and Suggestions for Authors

Comments 1: Check hyphoning throughout the manuscript.

Response: We sincerely thank the reviewers for their careful reading. We examined the entire manuscript and made revisions.

Comments 2: Check grammar in Figures 1 and 3.

Response: We sincerely thank the reviewer for careful reading. We checked and modified the syntax in Figure 1 and Figure 3, such as “fungi”.

Comments 3: Increase full size to gain visibility in Figures 3G, 5C-5D and 6C-6D

Response: We sincerely appreciate the valuable comments. We have tried our best to increase the size of Figures 3, 5 and 6.

We would like to thank the reviewer for professional review work, constructive comments and valuable suggestions on our manuscript.

Please do not hesitate to contact us if you have any questions. Thanks again.

Yours sincerely,

Jing Leng

Yunnan Agricultural University

lengjingrr@163.com

Reviewer 2 Report (New Reviewer)

Comments and Suggestions for Authors

This manuscript titled "Multi-omics reveals the effect of rumen microbiome and metabolome on milk fat contents in Binglangjiang buffaloes" presents an intriguing study addressing the relationship between rumen microbiota, host metabolites, and milk fat content in buffaloes. The use of multi-omics approaches enhances the novelty of this work, as it integrates microbiomics and metabolomics to uncover potential mechanisms. The findings provide insights into how microbiota-metabolite interactions influence milk fat synthesis.

1. Consider rephrasing the title to emphasize the novel multi-omics aspect and its implications more clearly (e.g., "Multi-omics Insights into Rumen Microbiota and Metabolite Interactions Regulating Milk Fat Synthesis in Buffaloes").

2. Abstract, please state the main conclusions more explicitly and include a sentence on the implications of the findings.

3. The background on milk fat synthesis and microbiota interaction is comprehensive. However, include recent references on multi-omics applications in dairy science. Clearly state the hypothesis and novelty of this study.

4. Provide details on quality control for LC-MS analysis and sequencing methods.

5. Figures 1–8 and supplementary tables are informative but need improved labeling for clarity (e.g., "HF vs LF" should be explicitly defined in captions).

6. The KEGG pathway enrichment results should explicitly indicate up- or down-regulated pathways in the HF and LF groups.

7. Discussion: Strengthen the discussion by integrating findings with existing literature on rumen microbiota’s role in fatty acid synthesis.

8. Highlight potential mechanisms by which microbiota, such as Quinella and Selenomonas, influence lipid metabolism.

9. Discuss implications for dairy production and future research directions (e.g., microbiota manipulation for milk quality enhancement).

Author Response

Response to Reviewers Comments

 Thank you very much for taking the time to review this manuscript. On behalf of all the contributing authors, I would like to express our deepest respect and gratitude to the editor and reviewers for your hard work. As the reviewer was concerned, there are several issues that need to be addressed. We tried our best to improve the manuscript and made some changes to the manuscript, the detailed corrections are listed below and marked in red in the revised manuscript. We are very grateful to the reviewer for their enthusiastic work and hope that the correction will meet with approval.

Comment: This manuscript titled "Multi-omics reveals the effect of rumen microbiome and metabolome on milk fat contents in Binglangjiang buffaloes" presents an intriguing study addressing the relationship between rumen microbiota, host metabolites, and milk fat content in buffaloes. The use of multi-omics approaches enhances the novelty of this work, as it integrates microbiomics and metabolomics to uncover potential mechanisms. The findings provide insights into how microbiota-metabolite interactions influence milk fat synthesis.

Response: We sincerely appreciate the valuable comments and constructive suggestions. We have carefully revised the article based on the reviewers' comments.

Point-by-point response to Comments and Suggestions for Authors

Comments 1: Consider rephrasing the title to emphasize the novel multi-omics aspect and its implications more clearly (e.g., "Multi-omics Insights into Rumen Microbiota and Metabolite Interactions Regulating Milk Fat Synthesis in Buffaloes").

Response: We sincerely appreciate the valuable comments and constructive suggestions. We carefully considered the reviewers' comments and revised the title to “Multi-omics reveals rumen microbiota and metabolite interactions that regulate milk fat synthesis in buffaloes”.

Comments 2: Abstract, please state the main conclusions more explicitly and include a sentence on the implications of the findings.

Response: We sincerely appreciate the valuable comments. In the abstract section, we have revised and restated the results and conclusions section, then briefly explained the significance of the findings of the study

Comments 3: The background on milk fat synthesis and microbiota interaction is comprehensive. However, include recent references on multi-omics applications in dairy science. Clearly state the hypothesis and novelty of this study.

Response: Thanks for your suggestion. As suggested by the reviewer, we have included some recent references on the application of multi-omics in dairy science in the introduction.

Comments 4: Provide details on quality control for LC-MS analysis and sequencing methods.

Response: We agree with the reviewer’s comments. In experimental methods, we describe the metabolomic LC-MS analysis and sequencing methods in more detail to make the methods used in this study clearer.

Comments 5: Figures 1–8 and supplementary tables are informative but need improved labeling for clarity (e.g., "HF vs LF" should be explicitly defined in captions).

Response: Thanks to the reviewer's suggestion. We have added the relevant description of the HF buffaloes in the caption of the figure.

Comments 6: The KEGG pathway enrichment results should explicitly indicate up- or down-regulated pathways in the HF and LF groups.

Response: We sincerely appreciate the valuable comments. As suggested by the reviewer, we indicated up- or down-regulated KEGG pathways in the annotation. A positive DA score indicates that the expression trend of all annotated differential metabolites in the pathway was upregulated in the HF group, and the length of the line segment indicates the absolute value of the DA Score, the size of the dots indicates the number of annotated differential metabolites in the pathway, The dots are distributed to the right of the center axis and the longer the line segment, the more the overall expression of the pathway tends to be up-regulated.

Comments 7: Discussion: Strengthen the discussion by integrating findings with existing literature on rumen microbiota’s role in fatty acid synthesis.

Response: We sincerely appreciate the valuable comments. The present study revealed differences in rumen microbial composition and host metabolites between buffaloes with high and low milk fat content. Among the microbiota that had a significant impact on these processes were Quienlla, Stilbella, Cyrenella, Microsphaeropsis and Paraphaeosphaeria. We have discussed all the microorganisms identified in the above studies, but their specific mechanisms in milk fat synthesis still need to be further investigated.

Comments 8: Highlight potential mechanisms by which microbiota, such as Quinella and Selenomonas, influence lipid metabolism.

Response: We sincerely appreciate the valuable comments. Previous research found that Quinella and Selenomonas all belong to the phylum Firmicutes. A higher relative ruminal propionate concentration was also found bin sheep with elevated populations of Quinella, this was hypothesized that Quinella would be conducive to lower methane production, and earlier studies had also supported that. There are also studies that have found Schwartzia is a genus in the Firmicutes, which was more abundant in cows with higher milk production. Lachnoclostridium and Lachnospiraceae UCG-006 could inhibit short-chain fatty acid-producing bacteria, thus inhibited the synthesis of milk fat, which was also confirmed by the significant enrichment of Lachnoclostridium in LF buffaloes in this study. However, the effects of microbiota such as Quinella and Selenomonas on lipid metabolism and their underlying mechanisms remain to be investigated, which is one of the directions of our future research.

 Comments 9: Discuss implications for dairy production and future research directions (e.g., microbiota manipulation for milk quality enhancement).

Response: The dominant rumen microbiota plays an important role in the lactation performance of the host, studies have shown that the relative abundance of some rumen microbiota has been found to have an effect on milk yield and milk fat content. The present study similarly found some microorganisms to be significantly enriched in the HF group. With the continuous development of sequencing technology, researchers have investigated the diversity of rumen microbes and their enzyme genes using techniques such as macrogenomics, macrotranscriptomics and macroproteomics. This opens the way for the discovery of more microfunctional microorganisms and enzyme genes in the future, but there are still a large number of unexploited microorganisms in the rumen, and their isolation and expression in engineered bacteria will hopefully improve milk fat synthesis.

We would like to thank the reviewer for professional review work, constructive comments and valuable suggestions on our manuscript.

Please do not hesitate to contact us if you have any questions. Thanks again.

Yours sincerely,

Jing Leng

Yunnan Agricultural University

lengjingrr@163.com

Reviewer 3 Report (New Reviewer)

Comments and Suggestions for Authors

Manuscript ID: animals-3380900

Title: Multi-omics reveals the effect of rumen microbiome and metabolome on milk fat contents in Binglangjiang buffaloes

General comment:

The manuscript explores an innovative and relevant topic in animal science and dairy production by employing a multi-omics approach to investigate the rumen microbiome and metabolome's role in shaping milk fat content in Binglangjiang buffaloes. This interdisciplinary methodology, combined with robust 16S rDNA and ITS sequencing, enhances the novelty and reliability of the research. The study provides comprehensive data on microbiome composition, metabolite profiles, and their interactions, supported by precise descriptions of sample collection, DNA extraction, and metabolomics analysis, ensuring reproducibility. The statistical analyses, including OPLS-DA and Spearman correlations, effectively link microbiota composition and metabolites to milk fat traits, offering credible evidence for the findings. The identification of microbial taxa and metabolites associated with higher milk fat content has significant practical implications for enhancing milk quality through microbiota modulation. With revisions to clarify findings, simplify visualizations, and incorporate recent literature, the manuscript has the potential to make a valuable contribution to the field. It is within the journal’s scope, and I would recommend its publication after addressing the outlined comments.

The manuscript would benefit from concise and visually appealing formats for figures and tables, with key findings highlighted using color coding or annotations. To enhance the discussion, recent studies published within the last two years should be cited to provide a more updated perspective on rumen microbiota and milk fat synthesis. Furthermore, discussing potential molecular mechanisms that link the identified microbial taxa and metabolites to milk fat synthesis would add depth to the analysis. The conclusion should clearly state the practical applications of the findings, such as dietary or microbial interventions to improve milk fat content. Lastly, thorough proofreading is essential to correct typographical errors and ensure compliance with the journal's formatting guidelines.

Specific comments:

Abstract and Title:

    • The abstract, while informative, lacks a clear statement of the study's objectives and key conclusions. Adding a specific summary of the findings would improve readability.
    • The title could be more concise while retaining specificity. For example, consider: "Multi-omics Analysis of Rumen Microbiome and Metabolome in Binglangjiang Buffaloes with Divergent Milk Fat Content."
  1. Introduction:
    • The introduction provides useful background but can benefit from more recent references on rumen microbiota and milk fat metabolism. For example, studies published after 2022 are underrepresented.
    • It would help to emphasize the novelty of studying Binglangjiang buffaloes compared to other buffalo breeds.
  2. Results:
    • The presentation of results is dense, particularly in the microbiome and metabolome sections. Tables and figures are helpful but could be streamlined for clarity. For instance, merging Figures 1D and 1E could reduce redundancy.
    • While the metabolic pathways enriched in high-fat (HF) buffaloes are well-explained, there is limited discussion on how these pathways might directly affect milk fat biosynthesis. Adding mechanistic insights or hypotheses would be beneficial.
    • The role of fungi in milk fat synthesis is highlighted but underexplored. A deeper analysis of fungal contributions, supported by relevant literature, could enhance the discussion.
  3. Discussion:
    • The discussion could more clearly distinguish between confirmed findings and speculative interpretations. For instance, the role of Quinella and Selenomonas in regulating milk fat synthesis should be backed by additional references or experimental data.
    • A comparison with similar studies on other buffalo or cattle breeds would provide a broader context for the findings.
  4. Language and Grammar:
    • The manuscript contains minor grammatical errors and awkward phrasing. For example, "Rumen metabolites may be directly involved in milk lipogenesis or indirectly affect milk lipogenesis..." could be rephrased for conciseness.
    • Avoid colloquial expressions like "as we all know" in the introduction.
  5. Figures and Tables:
    • Figures are informative but could be simplified. For instance, using heatmaps for microbiota-metabolite correlations would provide a more intuitive visualization.
    • Tables 1 and S3 lack a clear explanation of acronyms like DIM and DMI, which should be expanded in the table legends.

Author Response

Response to Editor Responses

Thank you very much for taking the time to review this manuscript. On behalf of all the contributing authors, I would like to express our deepest respect and gratitude to the editor for your hard work! As the reviewer was concerned, there are several issues that need to be addressed. We tried our best to improve the manuscript and made some changes to the manuscript, the detailed corrections are listed below and marked in red in the revised manuscript. We are very grateful to the reviewer for their enthusiastic work and hope that the correction will meet with approval.

Point-by-point response to Comments and Suggestions for Authors

Comments 1: Please check again the text. For example: Title of figure 1. Hylum level? atgenus?; Figure 3. fungiin? Analysisatphylum? Thegenus?; Line 290. Fat concents?; Line 535. Is it Binlangjian or Binglangjian?; Figure 5C. Plasma metabolome?

Response: We were really sorry for our careless mistakes. Thank you for your reminder. “Title of figure 1.  Hylum level? atgenus; Figure 3. fungiin? Analysisatphylum? Thegenus; Line 290. Fat concents; Line 535.  Is it Binlangjian or Binglangjian; Figure 5C.  Plasma metabolome?” have been changed to “Title of figure 1. Phylum level, at genus; Figure 3. Fungi, analysis at phylum, the genus; Line 323. fat contents; Line 498. Binglangjiang; Figure 5C. rumen metabolome”.

Comments 2: Material and methods

  • Lines 95-97. It is stated that rumen and blood samples were collected daily during 7 days for metabolomics analysis and even for VFA analysis. How was the day effect considered?
  • Line 97. Xue et al. (2018) is cited as reference for VFA analysis. These authors cited a Xue et al. (2007) and they in turn Hu et al. (2005). Please use the original reference.
  • How was dry matter intake recorded? Were the refusals analysed? Please provide this information.
  • To the best of my known, there are no publications on milk composition of Binglangjian buffalos in languages other than Chinese. It would be very useful to include as supplementary material a histogram of milk fat content using the 75 animals. This would allow readers to know the relevance of the low milk fat syndrome in Binglangjian buffaloes.

Response:

  • We sincerely thank the reviewers for their careful reading. We are sorry for the misunderstanding caused by our description, but we only collect rumen and blood samples on the same day, not on all 7 days.
  • Thank you for pointing this out. Based on your comments, we cited the original reference based on the reviewers' comments. Revised manuscript, line 117.
  • The feed intake and milk production of the buffaloes were recorded for 7 consecutive days prior to formal sampling, after which a 5-10% feed surplus was maintained. After recording the feed intake, the dry matter intake of the buffalo was calculated based on the measured dry matter content of the feed.
  • We think this is an excellent suggestion. Following the reviewer's suggestion, we have added a histogram of the milk fat content of 75 Binglangjiang buffaloes in the revised manuscript as been in Figure S1.

Comments 3: Results and discussion

  • The authors do not know whether animals with divergent milk fat also showed divergent feeding behavior and selected between dietary ingredients differently. This limitation should be mentioned in the discussion, as differences in microbiota could also be related to differences in feeding behavior.
  • The difference in milk fat content between divergent groups is greater than 275% (5.60 vs. 1.49%). Obviously, there may be some association between the ruminal microbiome and metabolome and milk fat composition, but considering that there were no differences between the groups in milk production and feed intake, this magnitude of difference suggests a genetic effect, more than differences in the ruminal microbiome and fermentation pattern. This possibility should be also mentioned in the discussion.
  • Different theories have been proposed to explain the phenomenon of milk fat depression in ruminants (see Dewanckele et al., 2020. J. Dairy Sc., 952, 6023). One of them is based on the antilipolytic effect of some intermediates of the biohydrogenation of fatty acids. However, it seems that no differences were found in any of these compounds in both the rumen and blood metabolome. Does this mean that this mechanism would be irrelevant in this case?
  • Line 298. It is stated that HF animals produce more fat synthesis precursors? Which one?
  • Line 361-362. It is claimed that LysoPE could increase milk fat. In addition to the study of Yamamoto et al. (2022) performed in hepatocytes, a more recent work (Wang et al., 2024. J. Dairy Sci. , 107) suggests the relationship between dairy fat and LysoPE.

Response:

  • We are grateful to the reviewer for comments, we have added the effect of differences in feeding behavior on rumen microbes to the talk section. Specifically as follows: “Meanwhile, changes in diet or feeding preferences can also lead to changes in rumen microbiota, which in turn have an impact on lactation performance. Exploring the mechanisms of rumen microbial metabolism of carbohydrates, proteins and sugars can be targeted to regulate rumen microbial fermentation and promote the production of milk component precursors through diets”. Revised manuscript, lines 301-305.
  • We sincerely appreciate the valuable comments and constructive suggestions. According to the editor's opinion, we have added the effect of genetic effects on milk fat synthesis to the talk section. Specifically as follows: “However, milk fat synthesis is the result of a concerted effort between gastrointestinal microbes and the organism. Not only do rumen microbes and host metabolites have an impact on milk lipid synthesis, but mammary lactation genes and signalling pathways also play a key role in the regulation of lipid synthesis. Therefore, it is necessary to further investigate the regulatory mechanism of milk lipid synthesis with the help of genomics and transcriptome, which is of great scientific significance to make full use of the lactation potential of buffaloes and produce high quality milk”. Revised manuscript, lines 417-424.
  • We feel great thanks for your professional review work on our article. The hydrogenation of unsaturated fatty acids from double bonds to single bonds on the carbon chain by rumen microorganisms is known as rumen microbial hydrogenation of unsaturated fatty acids, and some of the rumen hydrogenation intermediates do inhibit the formation of milk fat. However, there are more factors affecting biohydrogenation and the intermediates produced are more complex, and the absence of such differences in metabolites in this study does not mean that biohydrogenation does not exist. Quantitative analysis of microorganisms and metabolites affecting biohydrogenation is needed in future experiments.
  • Thank you for pointing this out. We have modified this to “Additionally, plant cell wall polysaccharide-degrading enzymes expressed by rumen microbiota were key in regulating the production of milk fat synthesis precursors, suggesting that HF buffaloes may be producing more milk fat synthesis precursors such as volatile fatty acids”. Revised manuscript, lines 329-332.
  • We sincerely thank the reviewer for careful reading. We added a description of the study by Wang et al. (Wang, M., Zhang, L., Jiang, X., Song, Y., Wang, D., Liu, H., Wu, S., & Yao, J. Multiomics analysis revealed that the metabolite profile of raw milk is associated with the lactation stage of dairy cows and could be affected by variations in the ruminal microbiota. J Dairy Sci, 107, 8709-8721(2024)). Revised manuscript, lines 396-398.

We sincerely thank the reviewers for their careful reading. We examined the manuscript and made revisions.

Please do not hesitate to contact us if you have any questions. Thanks again.

Yours sincerely,

Jing Leng

Yunnan Agricultural University

lengjingrr@163.com

This manuscript is a resubmission of an earlier submission. The following is a list of the peer review reports and author responses from that submission.

Round 1

Reviewer 1 Report

Comments and Suggestions for Authors

Dear Authors,

Your manuscript entitled "Multi-omics reveals the effect of rumen microbiome and metabolome on milk fat contents in Binglangjiang buffaloes" has serious flaws and, unfortunately, it might be considered not suitable for publication in Animals due to the lack of an appropriate experimental design for testing the factors under study and the parameters mentioned herein. The absence of any scientific criteria for running your experiment is an important limitation. No mention has been done to any scientific argument to justify how animals were assigned to each experimental group. No randomization has also been observed. No description of buffaloes (age, body weight, body condition scores, dry matter intake, etc.) has been given. No feed ingredients and nutritional value have been indicated for the diet tested. Furthermore, no experimental number approval has been provided. English needs revision for a native speaker. There are some evident grammatical and spelling errors throughout the manuscript that need attention. Figures' quality needs also to be improved to gain visibility. And, finally, objectives and conclusions need to be reviewed carefully.

Yours sincerely,

Reviewer.

Comments on the Quality of English Language

English needs revision for a native speaker. There are some evident grammatical and spelling errors throughout the manuscript that need attention.

Reviewer 2 Report

Comments and Suggestions for Authors

This is a well planned and very thorough investigation of associations between the rumen microbiome and the synthesis and composition of milk fat in buffaloes. This is important work that deserves to be published. There is, however, a major problem with the presentation in that the lettering in the complex text figures is so small that it needs to magnified about 5 times before it becomes possible to read. On the printed page, therefore, these figures will be incomprehensible.  Redesigning and redrawing these complex figures into a form that can be read  in the final text will not be easy but will be essential before this paper can be accepted for publication.